# Palladium-catalyzed asymmetric carbene coupling en route to inherently chiral heptagon-containing polyarenes

Huan Zhang[1,2], Chuan-Jun Lu[1,2], Gao-Hui Cai[1], Long-Long Xi[1], Jia Feng[1] & Ren-Rong Liu [1] ✉

Developing facile and direct synthesis routes for enantioselective construction of cyclic π-conjugated molecules is crucial. However, originate chirality from the distorted structure around heptagon-containing polyarenes is largely overlooked, the enantioselective construction of all-carbon heptagon-containing polyarenes remains a challenge. Herein, we present a highly enantioselective synthesis route for fabricating all carbon heptagon-containing polyarenes via palladium-catalyzed carbene-based cross–coupling of benzyl bromides and N-arylsulfonylhydrazones. A wide range of nonplanar, saddle-shaped tribenzocycloheptene derivatives are efficiently prepared in high yields with excellent enantioselectivities using this approach. In addition, stereochemical stability experiments show that these saddle-shaped tribenzocycloheptene derivatives have high inversion barriers.

Nonplanar π-conjugated molecules attract considerable attention owing to their intriguing electronic properties and tremendous untapped potential in materials science[1–3]. o-Tetraphenylene (tetrabenzo[a,c,e,g]-cyclooctatetraene) with a saddle-shaped structure is among the most interesting targets. These molecules exhibit an extraordinarily high barrier and scarcely undergo flipping, and their applications have been extensively studied[4,5]. A series of substituted tetraphenylenes have been employed as molecular devices, liquid crystals, and asymmetric catalysts.

Heptagon-containing polyarenes have various intriguing physical properties such as curved structures, unique one-dimensional packing, and promising nonlinear optics, conductivity, and magnetic properties (Fig. 1a)[6–13]. However, compared with the well-studied eight-membered o-tetraphenylene, originate inherent chirality from the distorted structure around heptagon-containing polyarenes is largely overlooked[14]. Although it was first characterized in tribenzocyclohepten-9-ylideneacetic acid by crystallization as a diastereomeric salt of brucine in 1965[15], the preparation of chiral heptagon-containing polyarenes has lacked studies for many decades. To date, very few tribenzocycloheptene derivatives could be

resolved into optical enantiomers. In 1967, Franke resolved a heteroanalogue tribenzothiepin-2-carboxylic acid (SO$_2$-heptagon) into optical enantiomers via its brucine salt. However, the resolution of tribenzoketone (CO-heptagon) was not successful[16]. Recently, Shibata reported the catalytic and enantioselective synthesis of multisubstituted tribenzothiepins and tribenzoselenepins via rhodium-catalyzed intermolecular cycloadditions[17]. However, the enantioselective construction of all-carbon heptagon-containing polyarenes that widely present in the materials remains unknown, and development of reliable and practical synthetic strategies for enantioselective synthesis of chiral heptagon-containing polyarenes is necessary.

In recent years, palladium-catalyzed carbene-based cross-coupling between N-tosylhydrazones and organohalides has become crucial for building carbon–carbon bonds[18–27]. However, these catalytic reactions are rarely explored in enantioselective syntheses and all the reported cases are around aryl halides (Fig. 1b). In 2016, Gu described the catalytic asymmetric synthesis of axially chiral multisubstituted alkenes from hydrazones and aryl halides[28]. Moreover, the N-tosylhydrazone-based carbene coupling reaction is a feasible approach for constructing

[1]College of Chemistry and Chemical Engineering, Qingdao University, Qingdao 266071, China. [2]These authors contributed equally: Huan Zhang, Chuan-Jun Lu. ✉e-mail: renrongliu@qdu.edu.cn

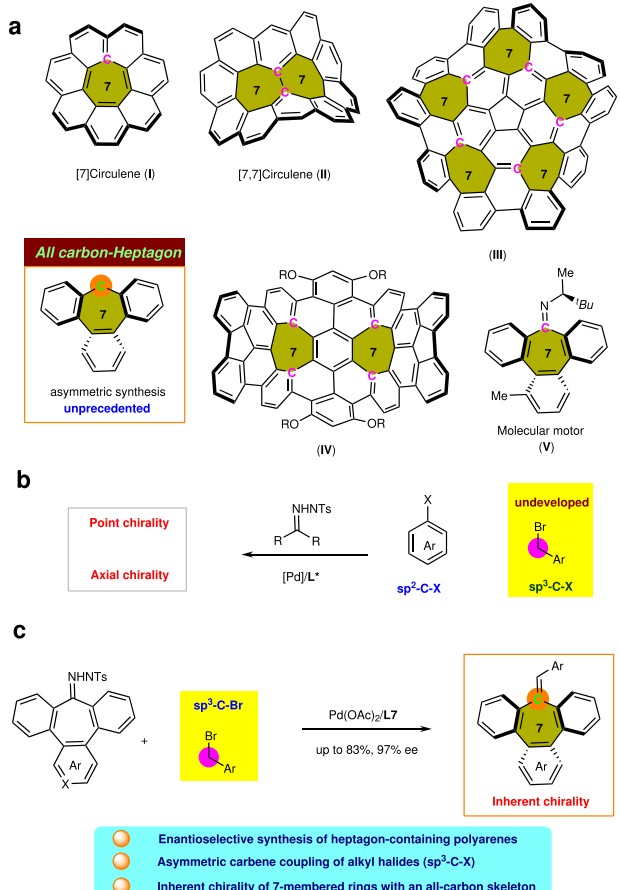

**Fig. 1 | Background and discovery. a** Heptagon-containing π-conjugated molecules. **b** Palladium-catalyzed asymmetric carbene coupling of N-tosylhydrazones and organohalides for the construction of point chirality and axis chirality. **c** Enantioselective synthesis of heptagon-containing polyarenes by palladium-catalyzed carbene-based couplings of unactivated benzyl halides.

two new C−X bonds with carbonyl functionality. Recently, Zhang proposed two palladium-catalyzed enantioselective three-component reactions of aryl bromides (N-sulfonylhydrazones) with organosilicon or alkynes for synthesizing chiral gem-diarylmethine silanes[29] or diarylmethyl alkynes[30] with good enantioselectivities. In these cases, the carbon−palladium intermediates formed via carbene migration cannot undergo β-hydrogen elimination and are captured by the nucleophiles to form the second C−X bond.

Compared to aryl halides, the benzylic electrophile is less explored in carbene-based couplings because benzyl electrophiles are often associated with slow oxidative addition of unactivated benzyl halides[31,32]. To date, the enantioselective carbene-based couplings of benzylic electrophile remain underexplored. We propose that β-hydride elimination of a quaternary C−Pd intermediate via migration-induced enantioselective insertion of a tribenzocycloheptene carbene leads to the formation of chiral tribenzocycloheptenes[33]. Herein, we report the palladium-catalyzed enantioselective synthesis of inherently chiral tribenzocycloheptene derivatives from benzyl bromides and tribenzotropone hydrazones (Fig. 1c).

## Results

### Reaction optimization
We initially reacted racemic N-arylsulfonylhydrazone **1a** with benzyl bromide **2a** as model substrates in the presence of 5 mol% of Pd(OAc)$_2$, 10 mol% of phosphoramidite ligand (**L1**), and lithium tert-

butanol in dioxane at 60 °C for 20 h. The reaction led to the formation of the chiral tribenzocycloheptene product **3a** with a yield of 42% and 24% ee (Table 1, entry 1). Several palladium catalysts were examined, and the yield and enantioselectivity were slightly improved when Pd$_2$(dba)$_3$ was used as the palladium source (entry 2). Similar results were obtained by employing [Pd(η-C$_3$H$_5$)Cl]$_2$ or Pd(dba)$_2$ as Pd sources (entries 3 and 4). Encouraged by this initial lead and previous palladium-catalyzed asymmetric carbene migratory insertion reactions[28], a series of chiral phosphoramidite ligands with different skeletons and substituents were examined (entries 5−12). The use of an isopropyl-substituted phosphoramidite ligand **L2** resulted in a significant increase in the enantioselectivity (67%), and the chemical yield also increased to 66% (entry 5). However, when the tetrahydropyrrole-substituted phosphoramidite **L3** was used as a ligand, the product was almost racemic (entry 6). Moreover, the enantioselectivity of **3a** could not be further improved by using the isopropyl-substituted spiro phosphoramidite **L4** as the ligand (entry 7). Notably, the yields and enantioselectivities obtained using the Feringa ligands (S, R, R)-**L5** and (S, R, R)-**L7** surpassed those obtained using **L2** (entries 8, 10). Further, the best yield and ee of 85% and 89%, respectively, were obtained using **L7** (entry 10). However, the 3, 3′-disubstituted phosphoramidite ligands (S, R, R)-**L6** and (R, R, R)-**L8** dropped the ee to 7% (entry 9) and 21% (entry 11), respectively. Notably, the yield and ee afforded by the H8-Feringa ligand (S, S, S)-**L9** were comparable with those obtained using (S, R, R)-**L7** (entry 12). Other solvents such as ether, tetrahydrofuran (THF), and toluene did not improve the yield and enantioselectivity of the product **3a** (entries 13−15). At low temperatures, the reactions resulted in products with high ee of 91% albeit with a slightly reduced yield (entry 16). Notably, the reduced palladium catalyst and ligand loading did not affect the enantioselectivity (entry 17). However, both the yield and ee sharply reduced when benzyl bromide was replaced with benzyl chloride under the standard conditions (entry 18). Therefore, entry 16 was selected as the best reaction condition for substrate scope investigation.

### Exploration of scope
With the determined optimized reaction conditions, we first evaluated the palladium-catalyzed carbene-based cross−couplings with a set of substituted benzyl bromides, and the corresponding results are shown in Fig. 2. The reaction displays an excellent functional group tolerance; for instance, alkyls (**3b**), aryls (**3c**), alkoxyls (**3t**), halogen (**3d−3f, 3p−3r**), CF$_3$ (**3k**), OCF$_2$X (**3h, 3l**), SCF$_3$ (**3m**), CN (**3g, 3s**), SO$_2$Me (**3i**), and CO$_2$Me (**3o**) can be presentposition of benzyl bromide and lead to the production of the corresponding tribenzo-cycloheptenes with good yields (62−83%) and excellent enantios-electivities (88−97% ee). Notably, sensitive functionalities, such as aldehyde (**3n**), aniline (**3u**), or sulfonamide (**3j**)-containing free hydrogens were also tolerated in the reaction. The introduction of two or three substituents at the para or meta of benzyl bromide (**3y−3ag**) only negligibly influenced the stereoselectivity and reactivity of the reaction. The tribenzocycloheptene products were achieved with yields of 54−67% and with 90−94% ee. X-ray crystallographic analyses of **3y** confirmed its absolute configuration. The reaction has obvious steric hindrance effect, and the enantioselectivity of the reaction decreases with the increase of ortho substituent group (**3v−3x**). Moreover, a variety of functional moieties could be embedded via the benzyl bromides. A tetraphenylethylene (**3aj**, an efficient fluorophore with aggregation-induced emission character-istics), biphenyl-triphenylmethyltetrazole (**3al**, olmesartan scaffold), the 2-cyanobiphenyl group (**3ai**, valsartan scaffold), or the cinnamate group (**3am**) could be embedded into the tribenzocycloheptene, and aromatic products with yields in the range of 45−80% yields and 88−93% ee were obtained. Notably, a series of heterocyclic compounds, such as pyridine (**3an, 3ao**), furan (**3aq**), and thiazole

**Table 1 | Pd-catalyzed carbene coupling between *N*-tosylhydrazone with benzyl bromide**

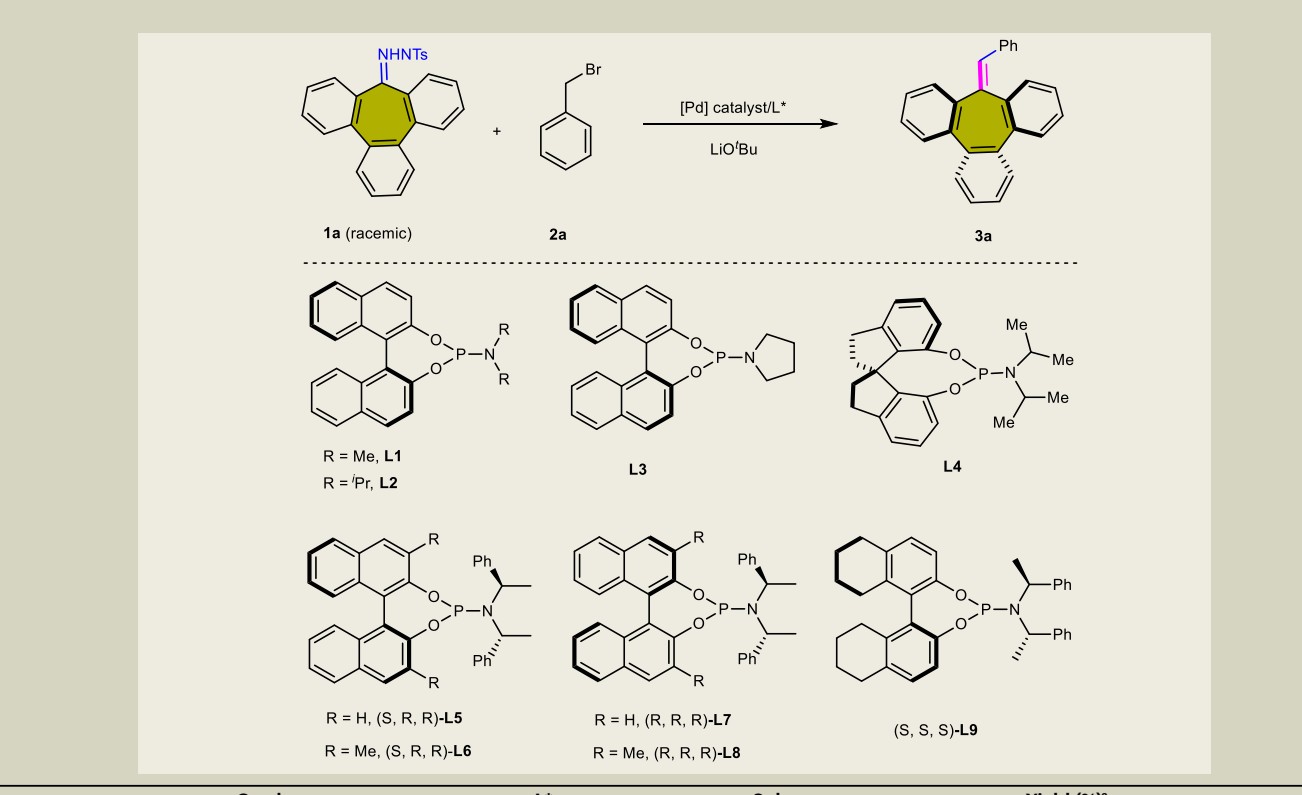

| Entry | Catalyst | L* | Solvent | Yield (%)[a] | ee (%)[b] |
|---|---|---|---|---|---|
| 1 | Pd(OAc)$_2$ | L1 | dioxane | 42 | 24 |
| 2 | Pd$_2$(dba)$_3$ | L1 | dioxane | 54 | 28 |
| 3 | [Pd(η-C$_3$H$_5$)Cl]$_2$ | L1 | dioxane | 40 | 14 |
| 4 | Pd(dba)$_2$ | L1 | dioxane | 50 | 24 |
| 5 | Pd$_2$(dba)$_3$ | L2 | dioxane | 66 | 67 |
| 6 | Pd$_2$(dba)$_3$ | L3 | dioxane | 57 | 5 |
| 7 | Pd$_2$(dba)$_3$ | L4 | dioxane | 55 | 62 |
| 8 | Pd$_2$(dba)$_3$ | L5 | dioxane | 78 | 77 |
| 9 | Pd$_2$(dba)$_3$ | L6 | dioxane | 55 | 7 |
| 10 | Pd$_2$(dba)$_3$ | L7 | dioxane | 85 | 89 |
| 11 | Pd$_2$(dba)$_3$ | L8 | dioxane | 52 | 21 |
| 12 | Pd$_2$(dba)$_3$ | L9 | dioxane | 78 | 89 |
| 13 | Pd$_2$(dba)$_3$ | L7 | Et$_2$O | 41 | 87 |
| 14 | Pd$_2$(dba)$_3$ | L7 | THF | 65 | 89 |
| 15 | Pd$_2$(dba)$_3$ | L7 | toluene | 40 | 87 |
| 16[c] | Pd$_2$(dba)$_3$ | L7 | dioxane | 80 | 91 |
| 17[d] | Pd$_2$(dba)$_3$ | L7 | dioxane | 68 | 90 |
| 18[e] | Pd$_2$(dba)$_3$ | L7 | dioxane | 42 | 50 |

Reaction conditions: the reaction was carried out at 0.1 mmol scale with palladium (5.0 mol%), **L*** (10.0 mol%), LiOtBu (3.0 equiv.), and the solvent (1.0 mL) in a sealed vial at 60 °C for 20 h, and the molar ratio of **1a**:**2a** was 1:1.2.
[a]Isolated yield.
[b]Determined by chiral HPLC.
[c]The reaction was performed at 50 °C for 36 h.
[d]Pd$_2$(dba)$_3$ (2.5 mol%), L7 (5.0 mol%), reaction time: 48 h.
[e]Benzyl chloride (0.12 mmol) was used instead of **2a** (2.5 mol%), **L7** (5.0 mol%), reaction time: 48 h.

(**3ap**) with different electronic properties, participated in the reaction without events; this result reaffirmed the robustness of this method. Importantly, in these cases (**3an**–**3ap**), benzyl bromide preferentially reacted with the bromine on the hetero-aromatic ring, and the bromine on the aromatic ring was retained after the reaction. This result suggests the possibility of constructing

more complex molecules through transition-metal-catalyzed cross-coupling reactions.

Inspired by the above results, we focused on expanding the generality of the tribenzocycloheptene N-arylsulfonylhydrazone **1**. As illustrated in Fig. 3, a wide range of *N*-arylsulfonylhydrazones were coupled with benzyl bromide **2a** to afford the desired heptagon-

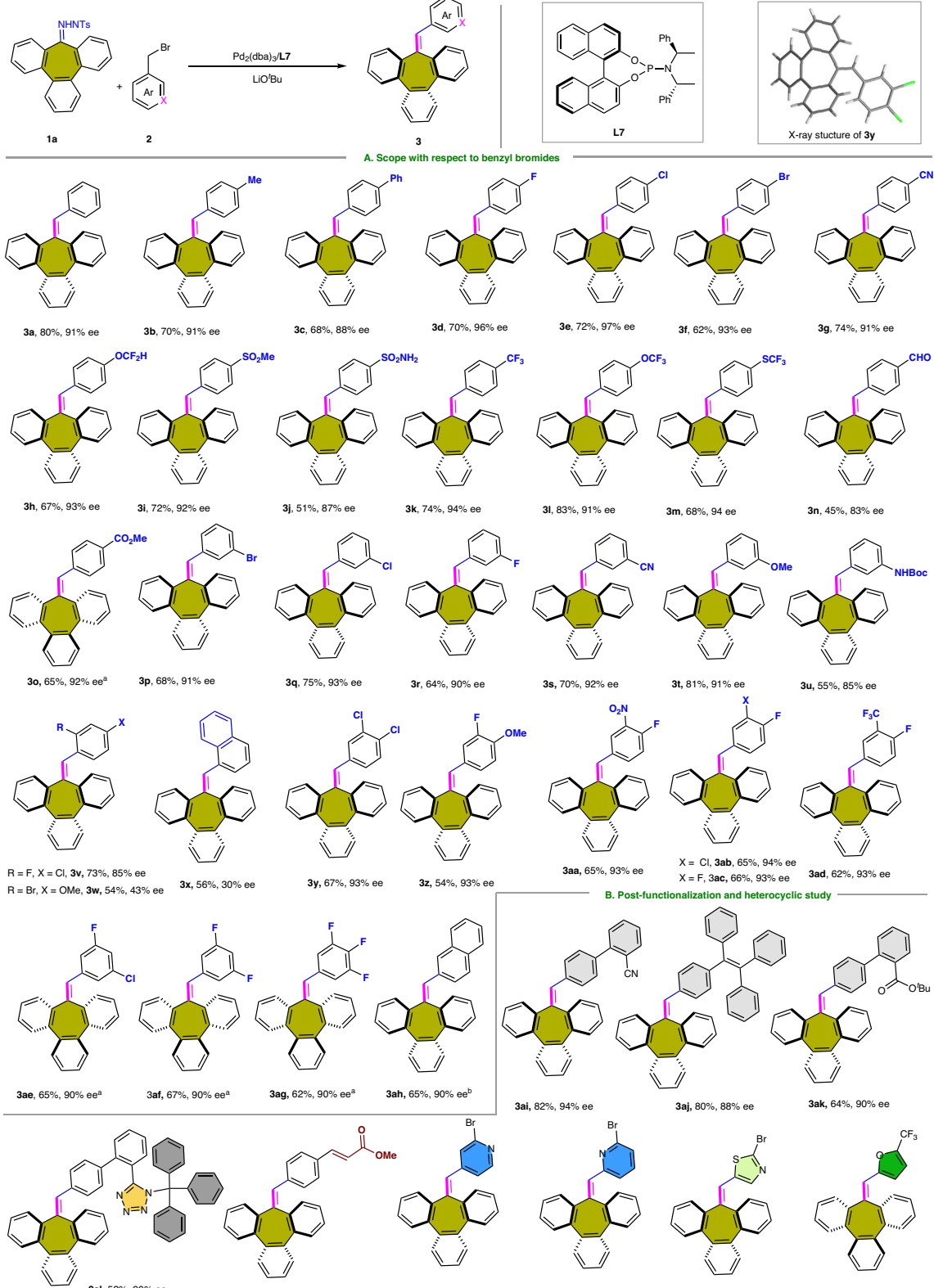

**Fig. 2 | Substituent effect of benzyl bromide substrate 2.** Reaction conditions: **1a** (0.2 mmol), **2** (0.24 mmol), Pd₂(dba)₃ (5 mol%), **L7** (10 mol%), LiOᵗBu (3.0 equiv.), dioxane (2.0 mL) in a sealed vial at 50 °C for 36 h. ᵃWith **L9** (10 mol%) as the ligand.

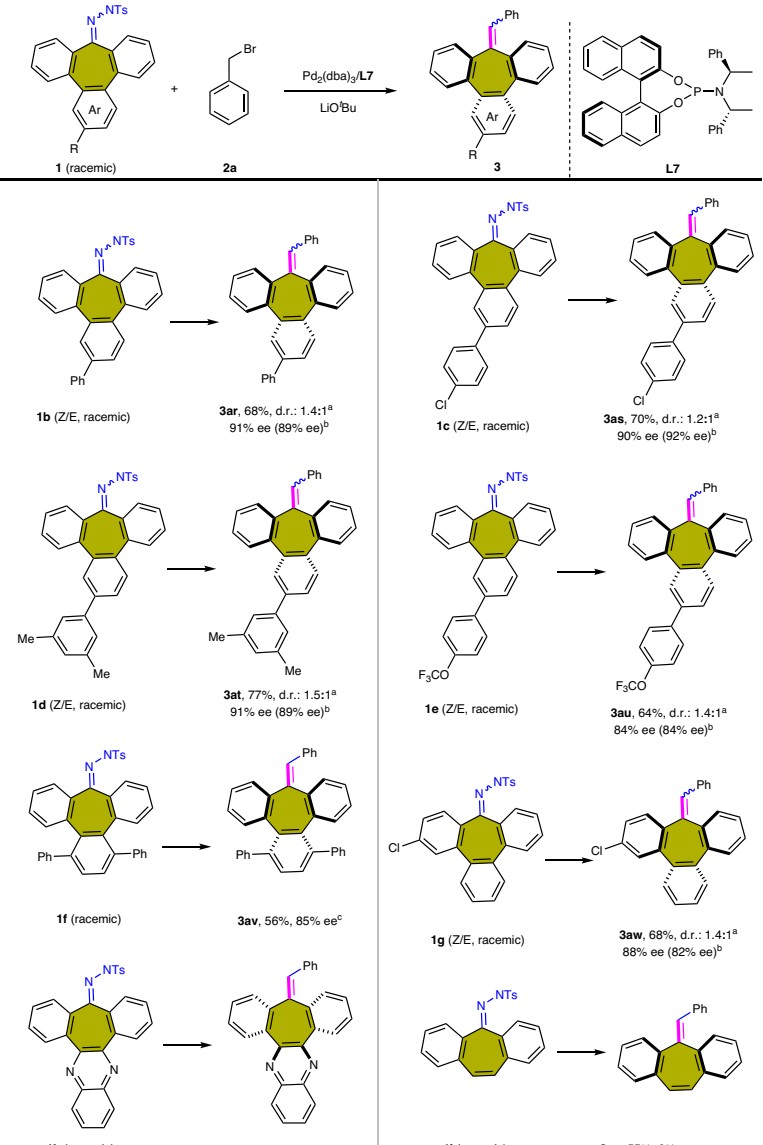

**Fig. 3 | Substituent effect of benzyl bromide substrate 2.** Reaction conditions: **1** (0.2 mmol), **2a** (0.24 mmol), Pd₂(dba)₃ (5 mol%), **L7** (10 mol%), LiOᵗBu (3.0 equiv.), dioxane (2.0 mL) in a sealed vial at 50 °C for 36 h. [a]The dr value represents the isomerism caused by the cis-trans of the phenyl group and the substituent R in the product. [b]The values in parentheses represent the enantioselectivity of the minor isomers. [c]With **L9** (10 mol%) as the ligand. d.r. values were determined by HPLC analysis.

containing polyarenes **3ar**–**3aw** with good yields and excellent enantioselectivities. To further investigate the scope of our reaction, we introduced heterocyclic rings into the tribenzocycloheptene skeleton to derive the desired product **3ax** with a yield of 58% and 80% ee. Although the hydrazone in dibenzosuberenone can also participate in the reaction, only the racemic product **3ay** can be obtained, which can be attributed to the low inversion barrier of the product.

**Synthetic utility and stereochemical stability study**
To assess the scalability and practicality of this palladium-catalyzed carbene-based cross-coupling strategy, the proposed approach was applied to several applications. The coupling reaction of N-arylsulfonylhydrazone **1a** with benzyl bromide **2a** was induced at a 2.0 mmol scale (Fig. 4a), and the product tribenzocycloheptene **3a** was obtained with 73% yield without loss of enantiopurity. The utility of the obtained products was demonstrated through palladium-catalyzed

Buchwald–Hartwig cross-coupling, Sonogashira coupling, and Suzuki coupling reactions, which provided the products **4**–**6** with high yields and enantioselectivities (Fig. 4b).

To demonstrate the stereochemical stability of these saddle-shaped tribenzocycloheptene compounds, we evaluated the racemization of **3a** at different times (Fig. 5a). Results of heating experiments indicated that the enantioselectivity of the compound **3a** decreased at a rate of approximately 1.3% per hour at 140 °C, and the calculated inversion barrier of **3a** was 31.7 kcal/mol (at 140 °C in xylene). Based on the Eyring-Polany equation, the steric hindrance on configurational stability was further demonstrated by the half-lives, in which the half-life of **3a** was 162 year at 298 K (see Supplementary information for details). Overcrowded alkenes represent molecular switches owing to the photoisomerization of the constituent C=C bonds[34]. The photostability of compound **3a** was then determined relative to the photoracemization rates in CH₃CN. The results showed that the half-life of **3a**, irradiated at 370 nm in a

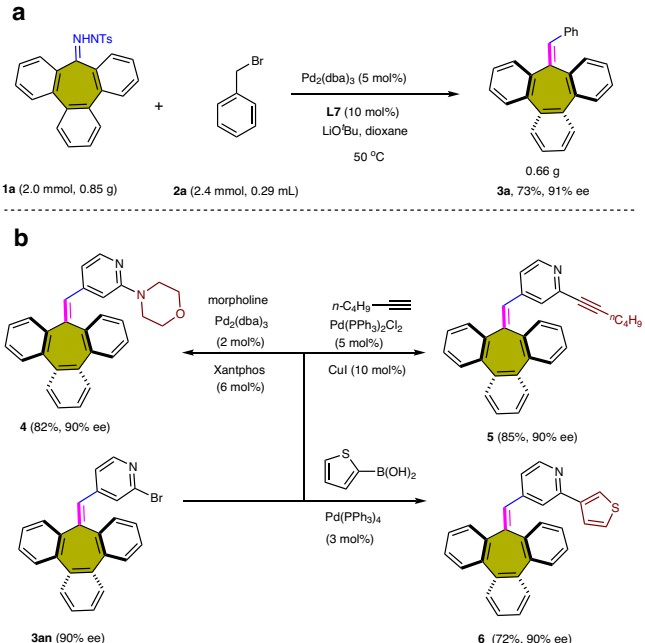

**Fig. 4 | Synthetic applications of palladium-catalyzed carbene-based cross-coupling. a** Gram-scale preparation of **3a**. **b** Further transformations with chiral tribenzocycloheptene compound **3an**. Representative tribenzocycloheptene product **3an** was shown to undergo Buchwald-Hartwig coupling, Sonogashira coupling and Suzuki coupling towards chiral heptagon-containing polyarenes **4**–**6**. These synthetic modifications occurred with excellent retention of optical purities.

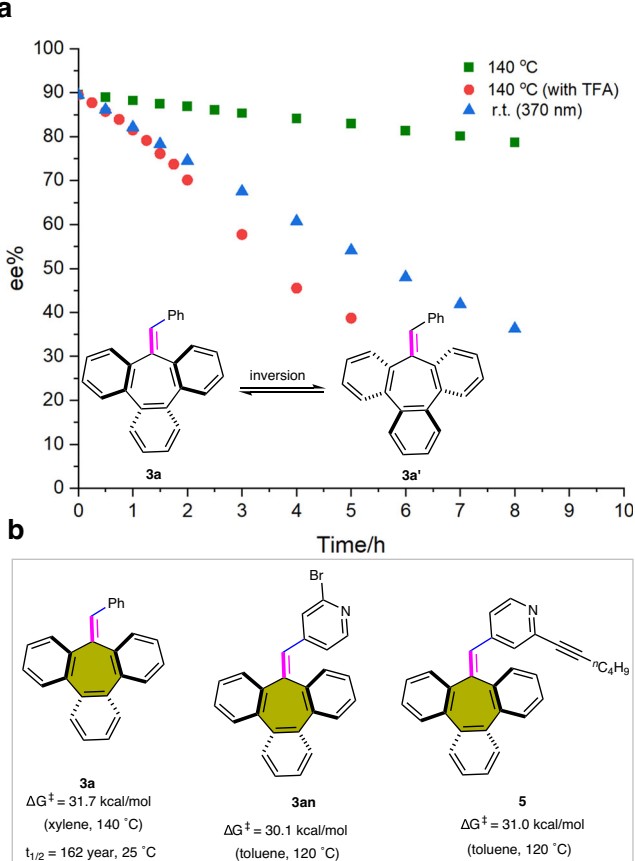

**Fig. 5 | Stereochemical stability studies of chiral tribenzocycloheptene compounds. a** Plot of ee value vs time under heating and light irradiated conditions. **b** The inversion energy barriers of representative saddle-shaped tribenzocycloheptene compounds **3a**, **3an**, and **5**.

photochemical reactor at ambient temperature, was ~6 h. Moreover, we also studied the stability of compound **3a** under acidic conditions, and in this case, the racemization of the compound is greatly accelerated. This may be due to the protonation of the cycloheptatrienylidene unit at the outer carbon atom could generate a stable tropylium cation, which could easily racemize since the newly generated exocyclic single bond. The inversion energy barriers of **3al** and **5** were also determined (Fig. 5b), and the values were 30.1 and 31.0 (120 °C) kcal/mol, respectively (see Supplementary information for details).

## Mechanistic investigation

Several control experiments were then conducted to gain insights into the reaction mechanism (Fig. 6). By simply changing the standard conditions, the partially saturated seven-membered ring product **7** captured by Ts can be obtained with a yield of 21% (Fig. 6, a1). However, under the standard conditions, **7** did not yield product **3a** by removing Ts groups (Fig. 6, a2). Moreover, diazo substrate **1a'** can obtain the comparable yield and enantioselectivity as **1a** under similar reaction conditions (Fig. 6, a3).

Based on the experimental results and previously reports[21], we proposed a mechanism as illustrated in Fig. 6b. The reaction is initiated with the oxidative addition of benzyl bromide **2** to a Pd(0) catalyst, a Pd(II) intermediate **A** is generated, which reacts with the diazo compound **1'** in situ generated from N-arylsulfonylhydrazone **1** in the presence of Li$^t$OBu, afforded the palladium–carbene species **C**. Alternatively, the palladium carbene intermediates **C** is generated by the coordination of Pd(0) with **1'**, which undergoes oxidative addition with **2** to furnish the Pd carbene intermediates **C**. The following carbene migratory insertion leads to the formation of a benzyl palladium species **E**, which then undergoes the β-hydride elimination process to deliver the final product **3** as well as regenerate the palladium catalyst with the aid of the base. In the

reaction, the ligand can also be exchanged first to form intermediate **D**, and then carbene migratory insertion to form **E**. The byproduct **7** is generated through the direct reductive elimination of the complex **E**.

## Discussion

In summary, a highly enantioselective synthesis route for heptagon-containing polyarenes was developed using a palladium-catalyzed carbene-based cross–coupling strategy. A wide range of chiral tribenzocycloheptene compounds were efficiently prepared with high yields and excellent enantioselectivities. Because of the ready availability of starting materials, high enantioselectivities, easy operability, and good functional group tolerance, the proposed method can be adopted as an efficient approach for preparing heptagon-containing polyarenes.

## Methods

### General procedure for the synthesis of chiral tribenzocycloheptene 3

Under nitrogen atmosphere, a mixture of **1** (0.2 mmol, 1.0 equiv.), **2** (0.24 mmol, 1.2 equiv.), Pd$_2$(dba)$_3$ (5 mol%), ligand (**L7** or **L9**, 10 mol%), LiO$^t$Bu (3.0 equiv.) in a Schlenk tube was added dioxane (2.0 mL). The resulting suspension was placed in an oil bath that had been preheated to 50 °C for 36 h. After completion of the reaction, the solvent was removed under vacuum and the crude product was purified with a chromatography column on silica gel to give **3** (petroleum ether/EtOAc).

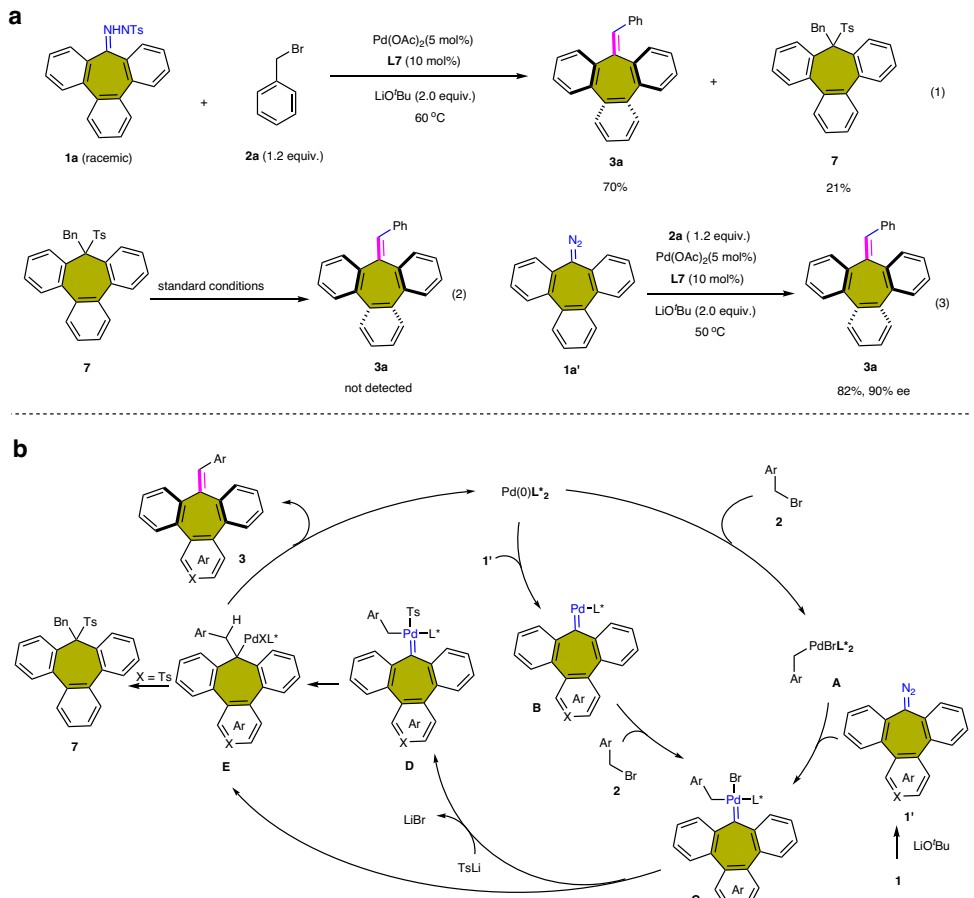

**Fig. 6 | Mechanistic studies. a** Control experiments. Reaction conditions for (1): **1a** (0.2 mmol), **2a** (0.24 mmol), Pd₂(dba)₃ (5 mol%), **L7** (10 mol%), LiO$^t$Bu (2.0 equiv.), dioxane (2.0 mL) in a sealed vial at 60 °C for 20 h. Reaction conditions for (2): **7** (0.1 mmol), Pd₂(dba)₃ (5 mol%), **L7** (10 mol%), LiO$^t$Bu (3.0 equiv.), dioxane (2.0 mL) in a sealed vial at 50 °C for 36 h. Reaction conditions for (3): **1a** (0.2 mmol), **2a** (0.24 mmol), Pd₂(dba)₃ (5 mol%), **L7** (10 mol%), LiO$^t$Bu (2.0 equiv.), dioxane (2.0 mL) in a sealed vial at 50 °C for 36 h. **b** Plausible catalytic cycle. β-Hydride elimination of a quaternary C-Pd intermediate.

## Data availability

Crystallographic data for the structures reported in this Article have been deposited at the Cambridge Crystallographic Data Centre, under deposition number CCDC 2302555 (**3w**). Copies of the data can be obtained free of charge via https://www.ccdc.cam.ac.uk/structures/. All other data supporting the findings of the study, including experimental procedures and compound characterization, are available within the paper and its Supplementary Information, or from the corresponding author upon request.

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

## Acknowledgements

Financial support from the Taishan Scholar Youth Expert Program in Shandong Province (tsqn201909096, R.R.L.), NSFC (22371152, R.R.L.) and Shandong Provincial Natural Science Foundation Project (ZR2023JQ006, R.R.L., ZR2022MB021, C.J.L.) are gratefully acknowledged.

## Author contributions

H.Z., C.J.L., G.H.C., L.L.X., and J.F. developed the catalysts and reactions and performed and analyzed experiments. H.Z. and C.J.L. performed the crystallographic studies; R.R.L. conceived and supervised the project and wrote the manuscript.

## Competing interests

The authors declare no competing interests.
