## [Peer Review File · Nature Communications]

Palladium-catalyzed asymmetric carbene coupling en route to inherently chiral heptagon-containing polyarenesReviewers' Comments:

Reviewer #1:

Remarks to the Author:

In this manuscript Liu and coworkers report an interesting enantioselective synthesis of inherently chiral heptagon-containing polyarenes using palladium-catalyzed asymmetric carbene coupling. The reaction conditions of the described transformation are simple and practical, and the scope shown in the manuscript is wide and diverse. A range of nonplanar, saddle-shaped tribenzocycloheptene derivatives were efficiently prepared in high yields with excellent enantioselectivities using this approach. In addition, the authors confirmed unequivocally the structure of the final products by single-crystal X-ray diffraction and performed an easy gram-scale synthesis and some follow-up reactions. Furthermore, through racemization experiments the authors have shown that these inherently chiral heptagon-containing polyarenes are relatively stable compounds, having a higher barrier for racemization. In conclusion, the manuscript features fantastic results in terms of reactivity and enantioselectivity, and provides an important advance of the state of the art of significance for the elaboration of novel challenging chiral scaffolds otherwise difficult to obtain. Therefore, it should be accepted for publication in Nature Communications after addressing the following minor remarks:

1. Authors should indicate the method of obtaining racemate 3 in the supporting information as a reference for HPLC analysis.
2. Screening of different bases for the reaction performed in dioxane may be interesting and could be added to the optimization table.
3. Regarding in Table 1, Pd(OAc)₂ is not placed correctly. It should be palladium, because it is referring all screened catalyst.
4. It would be nice to add one more example by variation of substituent at the side aryl in tribenzotropone.

Reviewer #2:

Remarks to the Author:

This manuscript by Liu et al describes the enantioselective synthesis of heptagon-containing polyarenes. The synthetic strategy is based on the cross-coupling of benzyl bromides and N-arylsulfonylhydrazones in a palladium-catalyzed reaction via carbene intermediates. The Pd-catalyzed cross-coupling of N-tosylhydrazones and organohalides is a very well-known strategy to create C-C bonds.

The use of N-tosylhydrazone as carbene precursor for coupling reactions is very well established. However, the use of benzyl halides is underexplored. The presented strategy is simple and applied in an efficient manner.

Some issues arised that should be clarified in the text or SI.

Firstly, the authors examined the palladium source, by testing only four different Pd complexes, the phosphoramidite ligand and also electrophile (benzylchloride was also tested) (table 1). Entry 16 was selected as optimized conditions (this should be clearly stated in the text).

- 1)-Have the authors try other Pd(0) source, such as Pd(PPh₃)₄? or Pd(II) such as PdCl₂(MeCN)₂?
- 2)-Why do the author choose the use of phosphoramidite ligands as chiral ligands? This is not commented.

Then, the authors studied the scope of the benzyl bromide in the reaction (Figure 2). Remarkably, a wide variety of substituents are allowed, showing the generality of the reaction together with good yields and enantioselectivities.

- 3)-Can benzyl tosylates or mesylates be used as electrophyles?

- 4)-Have the authors tried with ortho- substitution other that F (compound 3v)? I would encourage to try that and include the either positive or negative result in the manuscript.

The scope of the heptagon-containing N-arylsulfonylhydrazone was evaluated in Figure 3.

As described in SI, protocol A, compounds 1a to 1g where prepared from the corresponding ketones I.

- 5)-The synthesis of ketones I is not described in the manuscript and Angew. Chem. Int. Ed. 47,

7673–7676 (2008) is given as a reference. However, I could not find the synthesis of compound I in that reference. The synthetic protocol used for the preparation of all ketones I used as precursors for N-arylsulfonylhydrazones 1a to 1g should be described in the SI, together with appropriate references where required.

6)-Looking for heptagon-containing polyarenes, can further functionalized at the bottom part tribenzocycloheptene skeletons be used? I wonder if substitution in the 1 and 4 position is allowed (1,4-diphenyl-9H-tribenzo[a,c,e][7]annulen-9-one) as this should increase the inversion barrier. I encourage the authors to include that information in the manuscript.

The scalability of the reaction is remarkable (up to 660 mg of 3a obtained in a single reaction).

I also appreciate the application in subsequent synthetic steps that maintain the obtained ee, even when heating is required (i.e. heating at 100C for 12 h in the preparation of 4).

Regarding the stereochemical stability, 7)-I suggest to include the thermodynamic parameters of racemization by checking the rate of racemization at different temperatures and applying the extended version of Eyring-Polanyi equation. To be comparable with other systems, I suggest to estimate ΔG at 298 K

Overall, this manuscript is very well-developed. The supporting information is well accomplished. In my opinion, the enantioselective generation of heptagon containing polyarenes is novel and relevant. The reported results deserve publication after the above raised issues are addressed.

Reviewer #3:

Remarks to the Author:

The manuscript „Palladium-catalyzed asymmetric carbene coupling en route to inherently chiral heptagon-containing polyarenes“ submitted by the group of R.-R. Liu is a very interesting contribution to the field of polycyclic aromatic hydrocarbon chemistry. The authors have very thoroughly investigated the stereoselective transformation of tribenzocycloheptatriene hydrazones to tribenzocycloheptatrienes with exocyclic double bonds. They have studied mechanistic aspects of the reaction and have conclusively proven that by this way, stereochemically stable materials are obtainable. Although there is still a lot of work to do to reach a stereoselective synthesis of chiral nanographenes as shown in Fig. 1, III, this is a first and important step. I think this manuscript will find attention in the carbon-rich molecules community as well as have impact in the materials sciences. I recommend publication of this work in Nature Communication, but have some issues that should be addressed.

1. with regard to the stability, the authors have investigated the thermal and photochemical racemization of some of their compounds. I think it would give important information if the stability towards protons was studied. Protonation of the cycloheptatrienyliidene unit at the outer carbon atom could generate a stable tropylium cation, which could easily racemize due the newly generated exocyclic single bond.

2. In Fig. 1, the sp³-C-X compound in the yellow rectangles indicate a much broader scope of the reaction that was studied in the manuscript. I think a benzylic system should be used here to avoid confusion.

3. The authors present HRMS data of their compounds. Although the HPLC traces corroborate the purity of the compounds, I recommend that the full mass spectra together with a detailed look at the molecular ion peaks with calculated isomer distribution are shown.

Point-by-point responses to reviewer comments

Manuscript ID: NCOMMS-23-62253-T

Title: Palladium-catalyzed asymmetric carbene coupling en route to inherently chiral heptagon-containing polyarenes

Reviewer #1 (Remarks to the Author):

In this manuscript Liu and coworkers report an interesting enantioselective synthesis of inherently chiral heptagon-containing polyarenes using palladium-catalyzed asymmetric carbene coupling. The reaction conditions of the described transformation are simple and practical, and the scope shown in the manuscript is wide and diverse. A range of nonplanar, saddle-shaped tribenzocycloheptene derivatives were efficiently prepared in high yields with excellent enantioselectivities using this approach. In addition, the authors confirmed unequivocally the structure of the final products by single-crystal X-ray diffraction and performed an easy gram-scale synthesis and some follow-up reactions. Furthermore, through racemization experiments the authors have shown that these inherently chiral heptagon-containing polyarenes are relatively stable compounds, having a higher barrier for racemization. In conclusion, the manuscript features fantastic results in terms of reactivity and enantioselectivity, and provides an important advance of the state of the art of significance for the elaboration of novel challenging chiral scaffolds otherwise difficult to obtain. Therefore, it should be accepted for publication in Nature Communications after addressing the following minor remarks:

Our Response: We thank Reviewer 1 for agreeing with us on the importance of our work. We also appreciate all the helpful suggestions to improve the quality of our work. We have carefully revised our manuscript based on Reviewer 1's suggestions. Please see the following detailed points-to-points responses.

1. Authors should indicate the method of obtaining racemate **3** in the supporting information as a reference for HPLC analysis.

Our Response: Thanks for your suggestions. The method of obtaining racemate **3** for HPLC analysis was obtained by replacing Pd₂(dba)₃ and chiral ligand **L7** with a tetratriphenylphosphine palladium catalyst. We have added the method of obtaining racemate **3** in the supporting information.

2. Screening of different bases for the reaction performed in dioxane may be interesting and could be added to the optimization table.

Our Response: Thanks for your suggestions. Screening of different bases for the reaction, including K₃PO₄, Cs₂CO₃, K₂CO₃, NaOH, NaO^tBu was performed in dioxane. All the screening results were inferior to lithium tert-butanol. The result was added to the optimization table in the Supporting Information.

entry	base	solvent	yield (%)	ee (%)
1	K ₃ PO ₄	dioxane	66	75
2	Cs ₂ CO ₃	dioxane	68	55
3	K ₂ CO ₃	dioxane	62	77
4	NaOH	dioxane	70	34
5	NaO ^t Bu	dioxane	trace	-

3. Regarding in Table 1, Pd(OAc)₂ is not placed correctly. It should be palladium, because it is referring all screened catalyst.

Our Response: Thanks for your suggestions. We have revised it.

4. It would be nice to add one more example by variation of substituent at the side aryl in tribenzotropone.

Our response: We sincerely appreciate your suggestion. The variation of substituent at the side aryl group in tribenzotropone could also afford the desired product in good results (**3aw**, 68% yield, 1.4:1 dr, 88% ee/82% ee). The result was added in Fig. 2.

Reviewer #2 (Remarks to the Author):

This manuscript by Liu et al describes the enantioselective synthesis of heptagon-containing polyarenes. The synthetic strategy is based on the cross-coupling of benzyl bromides and N-arylsulfonylhydrazones in a palladium-catalyzed reaction via carbene intermediates.

The Pd-catalyzed cross-coupling of N-tosylhydrazones and organohalides is a very well-known strategy to create C-C bonds.

The use of N-tosylhydrazone as carbene precursor for coupling reactions is very well established. However, the use of benzyl halides is underexplored. The presented strategy is simple and applied in an efficient manner.

Some issues arised that should be clarified in the text or SI.

Firstly, the authors examined the palladium source, by testing only four different Pd complexes, the phosphoramidite ligand and also electrophyle (benzylchloride was

also tested) (table 1). Entry 16 was selected as optimized conditions (this should be clearly stated in the text).

Our response: Thanks for your suggestions. The optimal condition is clearly stated in the text.

1)-Have the authors try other Pd(0) source, such as Pd(PPh₃)₄? or Pd(II) such as PdCl₂(MeCN)₂?

Our response: We have tried other palladium sources with L1 as the ligand, such as Pd(PPh₃)₄, Pd(dmdba)₂ and PdCl₂(MeCN)₂, all the screening Pd(0) or Pd(II) sources were inferior to Pd₂(dba)₃. The result was added to the optimization table in the Supporting Information.

2) Why do the author choose the use of phosphoramidite ligands as chiral ligands? This is not commented.

Our response: The use of phosphoramides in the reaction was inspired by the previous palladium-catalyzed asymmetric carbene insertion reactions (ref. 28). In fact, during the conditional screening process, we also screened for other types of ligands, but the results were not as good as phosphoramides, and we now add these results to the optimization table in Supporting Information.

Then, the authors studied the scope of the benzyl bromide in the reaction (Figure 2). Remarkably, a wide variety of substituents are allowed, showing the generality of the reaction together with good yields and enantioselectivities.

3)-Can benzyl tosylates or mesylates be used as electrophiles?

Our response: Thanks for your comment. We tried benzyl tosylate or mesylate electrophiles, but no desired product **3a** was generated under the standard conditions.

4)-Have the authors tried with ortho- substitution other than F (compound **3v**)? I would encourage to try that and include the either positive or negative result in the manuscript.

Our response: Thanks for your comment. Other ortho- substitution of benzyl bromides were also examined. The results showed that the reaction has obvious steric hindrance effect, and the enantioselectivity of **3** decreases with the increase of ortho substituent group (**3w**, **3x**). These results were added in Fig. 2.

The scope of the heptagon-containing N-arylsulfonylhydrazone was evaluated in Figure 3. As described in SI, protocol A, compounds 1a to 1g were prepared from the corresponding ketones I.

5)-The synthesis of ketones I is not described in the manuscript and *Angew. Chem. Int. Ed.* 47, 7673–7676 (2008) is given as a reference. However, I could not find the synthesis of compound I in that reference. The synthetic protocol used for the preparation of all ketones I used as precursors for N-arylsulfonylhydrazones 1a to 1g should be described in the SI, together with appropriate references where required.

Our response: Thanks for your comment. We provided the references or synthetic protocol used for the preparation of all ketones I used as precursors for N-arylsulfonylhydrazones in the SI. 7-Membered cyclic ketones **Ii** is commercially available, **Ia**, **Ig** and **Ih** were known compounds and were synthesized following the reported procedures. **If** was synthesized following the reported procedure. Cyclic ketones **Ib-Ie** used in this study were prepared using reductive Heck and the subsequent dehydration reaction. We characterized all newly prepared ketones (**If**, **Ib-Ic**) and added their ¹H NMR, ¹³C NMR and HRMS to the Supporting Information. For **Ia**: Tochtermann, W.; Oppenlaender, K. & Walter, U. Seven-membered ring systems. II. Synthesis and rearrangement of tribenzocycloheptatrienone derivatives. *Chem. Ber.* **975**, 1329–1336 (1964).

For **Ig**: Zhou, L.; Sun, M.; Zhou, F.; Deng, Gu.; Yang, Y. & Liang, Y. Atmosphere-Controlled Palladium-Catalyzed Divergent Decarboxylative Cyclization of 2-Iodobiphenyls and α -Oxocarboxylic Acids. *Org. Lett.* **23**, 7150–7155 (2021).

For **Ih**: Chang, M.-Y.; Tsai, C.-Y. & Chan, C.-K. *m*CPBA-mediated conjugation of dibenzosuberone and amines or carboxylic acids. *Tetrahedron* **71**, 424–430 (2015).

For **If**: Pun, S. H.; Miao, Q. Introduction of Eight-Membered Rings to Polycyclic Arenes by Ring Expansion. *Chin. J. Org. Chem.* **40**, 3347–3353 (2020).

S1 was known compound and was synthesized following the reported procedures. *Chem. Ber.* 975, 1329–1336 (1964)

Under N₂ atmosphere, to a mixture of **S1** (5 mmol), **S2** (7.5 mmol), Pd(OAc)₂ (10.0 mol%), HCOONa (10 mmol) in DMSO (40 mL) was stirred at 100 °C using oil bath for 14 h. After the mixture was cooled to room temperature, water (10 mL) was added. The mixture was diluted with EtOAc (5 mL) and extracted with EtOAc (5 mL × 3). The combined organic layers were washed with brine (10 mL), dried (Na₂SO₄), and concentrated.

The above mixture (3 mmol), TsOH·H₂O (6 mmol) was stirred in toluene (30 mL) at 120 °C using oil bath. After completion of the reaction which was indicated by TLC, the reaction mixture was concentrated under reduced pressure. The residue was purified by silica gel column chromatography to obtain **I**.

2-Phenyl-9H-tribenzo[a,c,e][7]annulen-9-one (Ib) was synthesized by following Procedure A. The crude material was purified by column chromatography (SiO₂, petroleum ether: EtOAc = 20:1) to provide **Ib** as a white solid (0.57 g, 68% yield).

¹H NMR (400 MHz, CDCl₃) δ 7.88 (s, 1H), 7.79 (dd, *J* = 12.8, 7.8 Hz, 2H), 7.75 – 7.66 (m, 6H), 7.61 (tt, *J* = 7.8, 1.4 Hz, 2H), 7.49 (td, *J* = 7.1, 6.5, 3.9 Hz, 4H), 7.43 – 7.39 (m, 1H).

¹³C NMR (150 MHz, CDCl₃) δ 198.7, 143.0, 142.9, 141.1, 140.1, 137.1, 137.0, 136.8, 135.6, 131.7, 131.5, 129.8, 129.2, 129.0, 128.9, 128.2, 128.1, 127.8, 127.1, 127.0, 126.7, 126.6.

HRMS: (ESI) *m/z*: [M+H]⁺ Calcd for C₂₅H₁₇O 333.1274; Found 333.1271.

2-(4-Chlorophenyl)-9H-tribenzo[a,c,e][7]annulen-9-one (Ic) was synthesized by following Procedure A. The crude material was purified by column chromatography (SiO₂, petroleum ether: EtOAc = 20:1) to provide **Ic** as a white solid (0.62 g, 70% yield).

¹H NMR (400 MHz, CDCl₃) δ 7.94 – 7.74 (m, 3H), 7.73 – 7.68 (m, 3H), 7.66 (dd, *J* = 8.2, 2.0 Hz, 1H), 7.60 (t, *J* = 8.0 Hz, 4H), 7.52 – 7.43 (m, 4H).

¹³C NMR (150 MHz, CDCl₃) δ 198.6, 143.0, 142.9, 139.8, 138.5, 137.1, 136.9, 136.6, 135.9, 133.9, 131.8, 131.5, 129.6, 129.1, 129.1, 129.0, 128.3, 128.3, 128.2, 126.7, 126.7, 126.6.

HRMS: (ESI) *m/z*: [M+H]⁺ Calcd for C₂₅H₁₆OCl 367.0884; Found 367.0891.

2-(3,5-Dimethylphenyl)-9H-tribenzo[a,c,e][7]annulen-9-one (Id) was synthesized by following Procedure A. The crude material was purified by column chromatography (SiO₂, petroleum ether: EtOAc = 20:1) to provide **Id** as a yellow solid (0.61 g, 69% yield).

¹H NMR (400 MHz, CDCl₃) δ 7.86 (d, *J* = 1.4 Hz, 1H), 7.80 (ddd, *J* = 17.8, 7.9, 1.2 Hz, 2H), 7.72 (td, *J* = 4.1, 1.5 Hz, 4H), 7.62 (qd, *J* = 7.4, 1.5 Hz, 2H), 7.52 – 7.46 (m, 2H), 7.31 (s, 2H), 7.06 (s, 1H), 2.43 (s, 6H).

^{13}C NMR (150 MHz, CDCl_3) δ 198.7, 143.0, 142.9, 141.3, 140.0, 138.4, 137.2, 136.9, 135.4, 131.5, 131.5, 129.8, 129.4, 129.2, 129.0, 128.1, 128.0, 127.0, 126.6, 126.6, 125.0, 21.4.

HRMS: (ESI) m/z : $[\text{M}+\text{H}]^+$ Calcd for $\text{C}_{27}\text{H}_{21}\text{O}$ 361.1587 ; Found 361.1591 .

2-(4-(Trifluoromethoxy)phenyl)-9H-tribenzo[a,c,e][7]annulen-9-one (Ie) was synthesized by following Procedure A. The crude material was purified by column chromatography (SiO_2 , petroleum ether: EtOAc = 20:1) to provide **Ie** as a yellow solid (0.75 g, 60% yield).

^1H NMR (400 MHz, CDCl_3) δ 7.74 – 7.57 (m, 9H), 7.52 (t, J = 7.6 Hz, 2H), 7.44 – 7.38 (m, 2H), 7.25 (d, J = 8.2 Hz, 2H).

^{13}C NMR (100 MHz, CDCl_3) δ 198.5, 148.9, 143.0, 143.0, 139.6, 138.8, 137.1, 136.9, 136.6, 135.9, 131.8, 131.5, 129.8, 129.1, 129.0, 128.5, 128.3, 128.2, 126.8, 126.7, 126.6, 121.8, 121.3, 119.2.

HRMS: (ESI) m/z : $[\text{M}+\text{Na}]^+$ Calcd for $\text{C}_{26}\text{H}_{15}\text{F}_3\text{O}_2\text{Na}$ 439.0916; Found 439.0917.

1,4-Diphenyl-9H-tribenzo[a,c,e][7]annulen-9-one (If) was synthesized by known literature.³

^1H NMR (400 MHz, CDCl_3) δ 7.59 – 7.49 (m, 4H), 7.19 (q, J = 47.9, 28.5 Hz, 11H), 6.95 (dt, J = 15.1, 7.7 Hz, 5H).

^{13}C NMR (100 MHz, CDCl_3) δ 199.7, 145.7, 142.3, 142.1, 135.5, 134.2, 133.6, 131.2, 129.6, 128.9, 128.0, 127.5, 126.4, 124.7.

HRMS: (ESI) m/z : $[\text{M}+\text{H}]^+$ Calcd for $\text{C}_{31}\text{H}_{21}\text{O}$ 409.1587; Found 409.1593.

6)-Looking for heptagon-containing polyarenes, can further functionalized at the bottom part tribenzocycloheptene skeletons be used? I wonder if substitution in the 1 and 4 position is allowed (1,4-diphenyl-9H-tribenzo[a,c,e][7]annulen-9-one) as this should increase the inversion barrier. I encourage the authors to include that information in the manuscript.

Our response: Thanks for your suggestions. The variation of substituent at 1 and 4 position tribenzotropone (1,4-diphenyl-9H-tribenzo[a,c,e][7]annulen-9-one) could also afford the desired product in good results (**3av**, 56% yield, 85% ee). The result was added in Fig. 2.

The scalability of the reaction is remarkable (up to 660 mg of 3a obtained in a single reaction). I also appreciate the application in subsequent synthetic steps that maintain the obtained ee, even when heating is required (i.e. heating at 100 C for 12 h in the preparation of 4).

Our Response: Thanks for your suggestions. We have made a lot of attempts on the conversion of compound 3a under different conditions, but the results are not good. For example, under palladium-catalyzed conditions, it cannot react with substituted iodobenzene, and boric acid. Although the reaction can occur under high temperature conditions of strong acid, we cannot isolate the main compound as a pure product. Moreover, in the reaction with bromine, the enantioselectivity of the reaction decreased significantly.

Regarding the stereochemical stability, 7)-I suggest to include the thermodynamic parameters of racemization by checking the rate of racemization at different temperatures and applying the extended version of Eyring-Polany equation. To be comparable with other systems, I suggest to estimate DeltaG at 298 K

Our Response: Thanks for your suggestions. We performed heating experiments and calculations for the rotational energy barrier at different temperatures (393 K, 403 K, 423 K). Based on the Eyring-Polany equation, the steric hindrance on configurational stability was further demonstrated by the half-lives, in which the half-life of 3a was 162 years at 298 K.

Enantiomeric conversion half-life calculation

The Eyring Equation relates the activation free energy and rate constant:

$$k = \kappa \frac{k_B T}{h} e^{-\frac{\Delta G^\ddagger}{RT}} \quad (1)$$

(1) In this equation, ΔG^\ddagger is the Gibbs energy of activation, κ is the transmission coefficient, k_B is Boltzmann's constant, and h is Planck's constant. The transmission coefficient is often assumed to be equal to one as it reflects what fraction of the flux through the transition state proceeds to the product without recrossing the transition state.

The epimerization of atropoisomer is a first order reaction, which makes the half-life only relates to the reaction rate constant:

$$t_{1/2} = \ln(2)/k \quad (2)$$

The ΔG^\ddagger of **3a** at 298 K was estimate 30.9 kcal/mol. Based on Equations 1 and 2, we calculated the half-life of **3a**, $t_{1/2} = 162$ years.

Overall, this manuscript is very well-developed. The supporting information is well accomplished. In my opinion, the enantioselective generation of heptagon containing polyarenes is novel and relevant. The reported results deserve publication after the above raised issues are addressed.

Our Response: Thank you for agreeing with us on the importance of our work. We also appreciate all the helpful suggestions to improve the quality of our work.

Reviewer #3 (Remarks to the Author):

The manuscript “Palladium-catalyzed asymmetric carbene coupling en route to inherently chiral heptagon-containing polyarenes” submitted by the group of R.-R. Liu is a very interesting contribution to the field of polycyclic aromatic hydrocarbon chemistry. The authors have very thoroughly investigated the stereoselective transformation of tribenzocycloheptatriene hydrazones to tribenzocycloheptatrienes with exocyclic double bonds. They have studied mechanistic aspects of the reaction and have conclusively proven that by this way, stereochemically stable materials are obtainable. Although there is still a lot of work to do to reach a stereoselective synthesis of chiral nanographenes as shown in Fig. 1, III, this is a first and important step. In think this manuscript will find attention in the carbon-rich molecules community as well as have impact in the materials sciences. I recommend publication of this work in Nature Communication, but have some issues that should be addressed.

Our Response: Thank you for agreeing with us on the importance of our work. We also appreciate all the helpful suggestions to improve the quality of our work.

1. with regard to the stability, the authors have investigated the thermal and photochemical racemization of some of their compounds. I think it would give important information if the stability towards protons was studied. Protonation of the cycloheptatrienyliene unit at the outer carbon atom could generate a stable tropylium cation, which could easily racemize due the newly generated exocyclic single bond.

Our Response: Thanks for your suggestions. We studied the stability of compound **3a** under acidic conditions, and in accordance with your recommendation, under acidic conditions, the racemization of the compound is greatly accelerated. This may be due to the protonation of the cycloheptatrienyliene unit at the outer carbon atom could generate a stable tropylium cation, which could easily racemize since the newly generated exocyclic single bond.

2. In Fig. 1, the sp³-C-X compound in the yellow rectangles indicate a much broader scope of the reaction that was studied in the manuscript. I think a benzylic system should be used here to avoid confusion.

Our Response: Thanks for your suggestions. We have revised it.

3. The authors present HRMS data of their compounds. Although the HPLC traces corroborate the purity of the compounds, I recommend that the full mass spectra together with a detailed look at the molecular ion peaks with calculated isomer distribution are shown.

Our Response: Thanks for your suggestions. We added HRMS spectra of all the new compounds (**1b-1g**, **3a-3ay**) in the Supporting Information .

Reviewers' Comments:

Reviewer #1:

Remarks to the Author:

The authors have addressed most of the points raised. I would like to recommend the publication of this manuscript in Nature Communications without any delay.

Reviewer #2:

Remarks to the Author:

The authors have addressed or clarified all the raised issues. The revised manuscript includes all the required modifications and it has been considerably improved.

I recommend publication in Nature Communications

Reviewer #3:

Remarks to the Author:

The authors have addressed my concerns satisfactorily. I recommend publication of this manuscript in Nature communications.